# Glycation of Nail Proteins as a Risk Factor for Onychomycosis. Comment on Gupta et al. Diabetic Foot and Fungal Infections: Etiology and Management from a Dermatologic Perspective. *J. Fungi* 2024, *10*, 577

**DOI:** 10.3390/jof11010046

**Published:** 2025-01-08

**Authors:** Terenzio Cosio, Isabel Valsecchi, Roberta Gaziano, Elena Campione, Françoise Botterel

**Affiliations:** 1Department of Experimental Medicine, University of Rome Tor Vergata, Via Montpellier 1, 00133 Rome, Italy; roberta.gaziano@uniroma2.it; 2DYNAMYC UR 7380, Faculté de Santé, Université Paris-Est Créteil (UPEC), 94010 Créteil, France; isabel.valsecchi@u-pec.fr (I.V.); francoise.botterel@aphp.fr (F.B.); 3Dermatology Unit, Department of Systems Medicine, Tor Vergata University Hospital, 00133 Rome, Italy; elena.campione@uniroma2.it

**Keywords:** onychomycosis, diabetes, advanced glycation end-products

We read the review by Gupta et al. [1] published in the *Journal of Fungi* about diabetes-induced physical changes and their role in superficial fungal infections. We agree with Gupta’s overview and multifactorial pathogenesis, and we would like to comment on some putative direct fungal–host interactions and immunological views in this peculiar population.

In diabetes, hyperglycaemia causes a non-enzymatic glycation of free amino groups of proteins, known as the Maillard reaction, resulting in the formation of advanced glycation end-products (AGEs) [2]. AGEs are accumulated in tissues and the extracellular matrix (ECM) and they may act as mediators of diabetic complications. Within the ECM, AGEs, due to their ability to bind to different molecules, such as collagen, fibronectin, laminin, and elastin, alter the physiological properties of the matrix and increase its stiffness [2]. AGEs can also bind to the lysin residues of keratins, the major fibrillar proteins in skin and appendages, resulting in the gargantuan formation of N-linked glycoproteins on the cell membrane of keratinocytes and onychocytes.

In diabetic patients, AGE levels in the nail unit are higher than in healthy individuals, supporting the metabolic memory model, and this potentially contributes to the pathogenesis of onychomycosis [2,3]. In fact, AGEs form glycoproteins characterized by terminal mannose and galactose monomers, potentially acting as binding molecules for dermatophytes through a double mechanism, paving the way for a better understanding of the high prevalence of onychomycosis in diabetes (Figure 1). Adhesion is the first step in fungal infection. Conidia, the non-motile asexual spores of fungi, express adhesins on the cell walls. Adhesins are proteins that support the binding of the fungus to the epidermal matrix, especially fibronectin and other N-linked glycoproteins, and to the host keratinocyte membranes [4]. *Trichophyton* (*T.*) *rubrum,* and *T. mentagrophytes*, the leading causes of onychomycosis in diabetes, express carbohydrate-specific adhesins on the conidia surface that recognize mannose and galactose residues, which are abundantly expressed in human ECM and on glycated proteins [5,6]. As diabetic patients exhibit high levels of AGEs and glycated proteins in the nail unit, fungal conidia have a high possibility of binding to the cell surface and invading host tissues.

Furthermore, *T*. *rubrum* expresses proteins containing two or more lysin M (LysM) domains on the cell wall, called LysM proteins. Those proteins might simultaneously bind to fungal cell wall chitin or cell wall glycoproteins and to N-linked glycoproteins in humans [7]. In parallel with adhesins, LysM proteins could favour fungal adhesion to the host cell via overglycated proteins due to the high levels of AGEs in ECM, as well as on keratinocytes and onychocytes.

Moreover, as reported in the paper by Gupta et al. [1], hyperglycaemia also impairs immunoglobulin function through a non-enzymatic glycation process, supporting a potential role of adaptive immunity in response to fungal pathogens. However, an impairment in the antibody-mediated immune response could not be directly implicated in the pathogenesis of onychomycosis in people living with diabetes. In this regard, the proximal nail matrix (PNM) is considered a site of relative immunity, offering a relative safeguard against autoimmunity. In fact, in contrast with other regions of nail epithelium, both keratinocytes and melanocytes of the PNM have low expression in human leukocyte antigen (HLA)-A/B/C [8]. Furthermore, PNM exhibits unusually few CD1a(+) dendritic cells (DC_s_), and low numbers of CD4(+) and CD8(+) T lymphocytes, and natural killer and mast cells were found in the periungual nail matrix area. The low or absent expression of major histocompatibility complex (MHC) antigens in PNM, coupled with the reduced number of DC_s_, suggests a diminished antigen-presenting capability and, consequently, an impairment in adequate defence against fungal pathogens. Specifically, CD209/CD1a (+) DC_s,_ through the involvement of the mannose receptor (MR), can recognize the high-mannose-containing glycoproteins deriving from bacteria and fungi [8]. However, in a milieu characterized by high AGE levels, it is likely that these cells can bind favourably to the terminal mannoses of AGEs rather than to fungal cell wall antigens. Moreover, DC_s_, due to their crucial role as antigen-presenting cells, are at the interface between innate and adaptive immunity, coordinating the development of cell-mediated immune (CMI) response [9]. Considering, the low number of DC_s_ in PNM and the adaptive immunity disfunction in diabetes, fungal pathogens, such as *T. rubrum,* are more prone to infecting diabetic subjects and causing chronic infections.

Finally, potent immunosuppressive agents that are known as the key mediators of relative immune privilege, including transforming growth factor-beta (TGFβ)-1, insulin-like growth factor (IGF)-1, and the adrenocorticotropic (ACTH) hormone, are locally expressed in PNM more often than in the skin [8]. Interestingly, direct interactions between *T. rubrum* conidia and DC_s_ resulted in a downmodulation of MHC class II antigens and favoured the production of IL-10 [9], enhancing local immunotolerance and leading to a reduction in host defence mechanisms against fungal infection.

Thus, nail immune privilege, in concert with the high AGEs nail levels, can prejudice effective anti-fungal immune responses, increasing the risk of mycoses.

In this light, the bivalent role of glycated proteins in host–pathogen interactions suggests the need for further research on these pleiotropic molecules and their role in diabetes-related onychomycosis.

## Figures and Tables

**Figure 1 jof-11-00046-f001:**
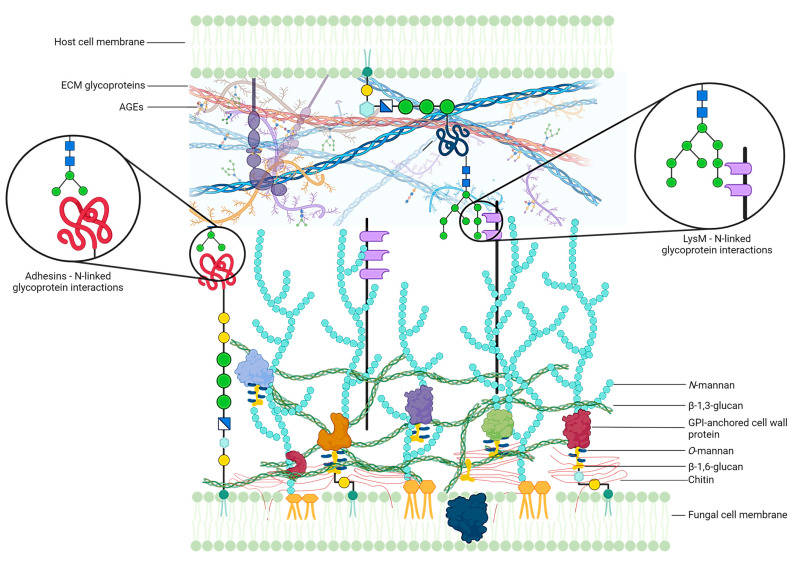
Putative interactions of fungi with the human host cells or extracellular matrix (ECM). In diabetic patients, a high level of AGEs has been described in the ECM, which form glycoproteins through the non-enzymatic glycation processing of proteins. Adhesins bind to mannose and galactose residues, which are expressed on AGEs in the ECM and on glycosylated proteins of the host’s cell membrane. LysM proteins, which are expressed on the fungal cell wall, could bind to fungal cell wall chitin, cell wall glycoproteins, and N-linked glycoproteins in human skin. Created in BioRender.com (accessed on 23 October 2024).

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
