# Peer review of "Glycation of Nail Proteins as a Risk Factor for Onychomycosis. Comment on Gupta et al. Diabetic Foot and Fungal Infections: Etiology and Management from a Dermatologic Perspective. J. Fungi 2024, 10, 577"

_jof, 2025, doi:10.3390/jof11010046_

Round 1
Reviewer 1 Report
Comments and Suggestions for Authors
Although the aforementioned paper by Gupta et al was a general review of the topic, mainly focusing on clinical aspects, I find that the Comment, focusing on mechanisms of fungus-host interaction, clarifies the putative mechanisms behind the increased incidence of tinea unguium in diabetics
Line 41-43, “In particular, Trichophyton (T.) rubrum and T. mentagrophytes, the leading cause of onychomycosis in diabetes, express carbohydrate-specific adhesins on conidia surface that recognise mannoses and galactoses residues, abundantly expressed in human ECM and on glycated proteins [5,6].”; as T. interdigitale is the second more common dermatophyte in tinea unguium and the mentioned “T. mentagrophytes” is actually T. quinckeanum (the mentioned strain in ref. 5 was ATCC 11481, today classified as T. quinckeanum), “T. mentagrophytes” should be deleted from the sentence.